# Artificial Intelligence-Enabled Retinal Vasculometry at Scale Utilizing the UK Biobank, CLSA, and NEL DESP Datasets

Roshan A. Welikala[1], Jiri Fajtl[1], Gordon Johnson[1], Farzana Rahman[1], Razvan Podoleanu[1], Paolo Remagnino[2], Ellen E. Freeman[3], Ryan Chambers[4], Louis Bolter[4], John Anderson[4], Abraham Olvera-Barrios[5], Alasdair Warwick[5], Paul J. Foster[5], Royce Shakespeare[6], Rahul Ganguly[6], Catherine Egan[5], Adnan Tufail[5], Christopher G. Owen[6], Alicja R. Rudnicka[6], Sarah A. Barman[1], on behalf of the UK Biobank Eye and Vision Consortium, on behalf of the ARIAS Research Group

[1] School of Computer Science and Mathematics, Kingston University, London, UK
[2] Department of Computer Science, Durham University, UK
[3] School of Epidemiology and Public Health, University of Ottawa, Ottawa, Canada
[4] Diabetes and Endocrinology, Homerton Healthcare NHS Foundation Trust, London, UK
[5] NIHR Biomedical Research Centre, Moorfields Eye Hospital NHS Foundation Trust and UCL Institute of Ophthalmology, London, UK
[6] Population Health Research Institute, St George's University of London, London, UK

*Abstract*—Retinal imaging offers a non-invasive means to assess the circulatory system, with morphological features of retinal vessels serving as biomarkers for systemic disease. QUARTZ (QUantitative Analysis of Retinal vessel Topology and siZe) is a fully automated artificial intelligence-enabled retinal vasculometry system designed to process large-scale retinal image datasets to obtain quantitative measures of vessel morphology for use in epidemiological studies. Previously reliant on traditional image processing and machine learning, QUARTZ has now transitioned to a deep learning pipeline. Currently individually trained versions are tailored to specific datasets. Evaluation using the UK Biobank retinal dataset shows improvements in performance metrics: the F1 score for vessel segmentation increased from 0.7753 to 0.8472, accuracy for the A/V segment-level decision increased from 0.8524 to 0.9022, the detection rate for optic disc localization increased from 0.9760 to 0.9933, and the F1 score for image quality classification increased from 0.8872 to 0.9750. QUARTZ distinguishes itself from other deep learning based retinal vasculometry systems through its efficient use of data, extracting valuable information despite issues such as low levels of illumination. The high performance of QUARTZ is consistent across two other extensive retinal datasets, namely the Canadian Longitudinal Study on Aging (CLSA) and the North East London Diabetic Eye Screening Programme (NEL DESP). Evaluation on subsets was preceded by the automatic processing of entire retinal datasets by QUARTZ, processing over 1.4 million images. These retinal vasculometry outputs will serve as a valuable resource for epidemiological studies.

*Keywords*—Retinal Vasculometry, Deep Learning, Artificial Intelligence, Epidemiological Studies, UK Biobank, CLSA, NEL DESP.

## I. INTRODUCTION

Examination of retinal images offers a direct and non-invasive view of the blood circulatory system. The morphological characteristics of retinal vessels, including width and tortuosity, have been prospectively associated with systemic disease [1], [2]. Therefore, the eye can be considered a window to the health of the body, providing biomarkers for risk prediction not only of ocular disease (e.g., glaucoma and diabetic retinopathy) but also systemic disease such as diabetes and cardiovascular disease, which includes heart attack and stroke [3].

QUARTZ (QUantitative Analysis of Retinal vessel Topology and siZe) is a fully automated artificial intelligence-enabled retinal vasculometry system developed by our research group. QUARTZ has previously been used to process large retinal image datasets from the UK Biobank [4], [5], EPIC-Norfolk [6], [7], and FOREVER [8], [9] cohorts to obtain quantitative measures of vessel morphology. This has contributed to many epidemiological studies [3], [10], [11], [12], [13], [14] and recently demonstrated that risk scores to predict circulatory mortality, heart attack, and stroke derived using retinal vasculometry performed similarly to established risk scores [3]. Whilst end-to-end deep learning disease prediction models continue to gain prominence [15], [16], [17], [18], retinal vasculometry is still a very active methodology as it offers interpretable results, identifying specific vascular features and changes which predict disease status.

Whilst artificial intelligence has already been integrated into QUARTZ via deep learning [5], [19], the system heavily relied on traditional image processing and machine learning. Deep learning has now been extended to all core modules of QUARTZ. Other comparable systems exist, which include VAMPIRE [20], AutoMorph [21], RMHAS [22], some already presenting full deep learning pipelines. However,

This research has been conducted using the UK Biobank resource under application number 522. The UK Biobank Eye and Vision Consortium is supported by funding from The Special Trustees of Moorfields Eye Hospital NHS Foundation Trust, and at the NIHR Biomedical Research Centre at Moorfields Eye Hospital and UCL Institute of Ophthalmology.

This research was made possible using the data/biospecimens collected by the Canadian Longitudinal Study on Aging (CLSA). Funding for the Canadian Longitudinal Study on Aging (CLSA) is provided by the Government of Canada through the Canadian Institutes of Health Research (CIHR) under grant reference: LSA 94473 and the Canada Foundation for Innovation, as well as the following provinces, Newfoundland, Nova Scotia, Quebec, Ontario, Manitoba, Alberta, and British Columbia. This research has been conducted using the CLSA Baseline Comprehensive Dataset version 7.0 and Follow-up 1 Comprehensive Dataset version 4.0 under Application Number 2209017. The CLSA is led by Drs. Parminder Raina, Christina Wolfson and Susan Kirkland. The opinions expressed in this manuscript are the author's own and do not reflect the views of the Canadian Longitudinal Study on Aging. Data are available from the Canadian Longitudinal Study on Aging (www.clsa-elcv.ca) for researchers who meet the criteria for access to de-identified CLSA data.

Image analysis work using the CLSA data was funded by Canadian Institutes of Health Research Award (PJT183690).

Collection of data from the Homerton Healthcare NHS Foundation Trust and image analysis work was funded by the Wellcome Trust Collaborative Award (224390/Z/21/Z).

Corresponding author: Roshan Welikala (r.welikala@kingston.ac.uk).

QUARTZ remains distinct as it is geared towards epidemiological studies, aiming to maximise useful data extracted from large cohort studies which can include larger amounts of poorer quality images. For instance, QUARTZ can effectively utilise partially illuminated images by extracting information from well segmented sections of the vasculature. Hence, all deep learning modules in QUARTZ have been trained and evaluated on a diverse range of images, including those with low levels of illumination. Whereas other systems [21], [22] select images for analysis based on the EyeQ dataset [23], which is specific to lesion detection and requires the main structures and lesions to be clear enough to provide a diabetic retinopathy grade.

In this paper, using UK Biobank, the performance of the core modules of the latest version of QUARTZ is presented, demonstrating a shift from the previous version [5] to a deep learning pipeline. Additionally, the performance of QUARTZ is presented for two other large retinal image datasets from the Canadian Longitudinal Study on Aging [24] and the North East London Diabetic Eye Screening Programme [25]. Following training and evaluation on subsets, QUARTZ was used to automatically process the entire retinal datasets. A generalised system [21], [22], trained across multiple retinal datasets would offer broader application. However, our current focus is optimising performance, so a separate version of QUARTZ has been tailored to each dataset. This approach will create a performance baseline to enable comparative analyses into the advantages and disadvantages of transitioning QUARTZ towards a generalised system.

## II. MATERIALS

UK Biobank (UKBB) [4] is a large prospective cohort study for which baseline biomedical and physical assessments were carried out in 2006–2013, in 502,682 adults aged 40–69 years recruited from 22 UK centres. During 2009-2013, a subset of 85,746 participants had retinal images captured, providing 175,856 images. Colour images were captured with the Topcon 3D-OCT 1000 Mark 2 fundus camera. Images were non-mydriatic, macular centred, from both eyes, with a 45° field-of-view (FOV), and saved in PNG format with a resolution of 2048 x 1536 pixels. The UKBB study was approved by the Northwest Region NHS research ethics committee.

The Canadian Longitudinal Study on Aging (CLSA) [24] is a large long-term cohort study for which baseline biomedical and physical assessments were carried out in 2012-2015 in 30,097 adults aged 45-85 years recruited from 11 sites (in 7 Canadian provinces). Participants were seen every 3 years. From baseline and the first follow-up examinations, 29,635 participants had retinal images captured, providing 106,506 images. Of which, 24,160 participants had images captured from both examinations. Age-related eye disease is evident in the dataset due to the participant age range. Colour images were captured with the Topcon TRC-NW8 fundus camera with the Nikon D90 camera attached. Images were non-mydriatic, macular centred, from both eyes, with a 45° FOV, and saved in JPEG format with resolutions of 4288 x 2848 and 4928 x 3264 pixels. All images were resized to the most common resolution of 4288 x 2848 pixels, simplifying future pixel-micron conversions. The CLSA study was approved by 13 research ethics boards across Canada.

The North East London Diabetic Eye Screening Programme (NEL DESP) [25] is based at the Homerton Healthcare NHS Foundation Trust and offers annual diabetic eye screening to a large ethnically diverse (with high representation of white, South Asian, and black individuals) population with diabetes, with a spectrum of diabetic eye disease and a wide age range. An initial dataset was made available for training and evaluating QUARTZ, consisting of 1000 expired screening encounters, providing 6,268 images. A larger dataset was curated from 202,886 consecutive routine screening encounters between 1st January 2021 and 31st December 2022, resulting in over 100,000 patients and 1,175,423 images. Patient IDs were pseudonymised. Colour images were captured from a range of fundus cameras including models from Canon and Topcon. Images were mydriatic, macular centred and optic disc centred, from both eyes, with a 45° FOV, and saved in JPEG format with various resolutions ranging from 150 x 300 to 6000 x 4000 pixels. Non-retinal images (e.g., crystalline lens, eyelids, hands) were included as part of the usual protocol to document anterior segment pathology or to confirm camera functioning when the patient could not be photographed. All images were resized to the second most common resolution of 3648 x 2432 pixels, simplifying future pixel-micron conversions. The study was approved by the NHS Health Research Authority, although full research ethics approval was not required as all data were pseudonymised.

## III. METHOD

QUARTZ was structured into the core modules of arteriole/venule segmentation, optic disc/cup segmentation, image quality classification, and vessel analysis (see Fig. 1). Algorithm details of the first three modules are described in this section, each designed with a novel approach. Separate models were trained for each of the three datasets; the training hyperparameters listed in this section were for the UKBB dataset. The vessel analysis module, which included measurements of width and tortuosity, used previous well documented and validated techniques [5].

### A. Pre-processing

Colour images were cropped to FOV, and then adjusted to achieve a square shape by either centrally cropping or evenly zero padding the vertical dimension. Zero padding was particularly useful for the NEL DESP dataset which included some images with the circular FOV truncated at the top and bottom.

### B. Arteriole/Venule Segmentation

For performing arteriole/venule (A/V) segmentation the U-Net architecture [26] was used as a starting point, which is a convolutional neural network (CNN) that was initially proposed for biomedical image segmentation. The U-Net was adapted to employ a multi-segmentation technique [27] where the network outputs three independent channels to generate separate binary segmentation maps for the structures of arterioles, venules, and vessels. This configuration allowed vessel crossings to be handled more intuitively, avoiding commonly used output pixel labels of crossing and uncertain. The U-Net was further adapted using ConvNeXt blocks [28], which followed from ConvNeXts [29] (a family of pure CNN models) having been previously demonstrated to compete favourably with Transformers [30].

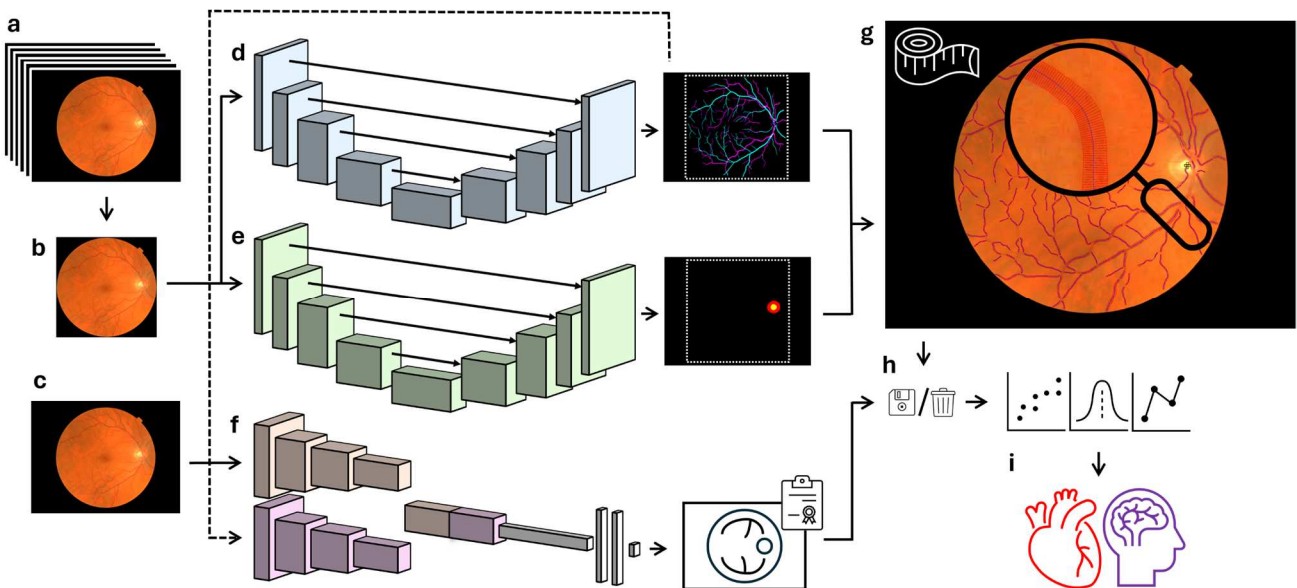

Fig. 1. The use of QUARTZ, from dataset to biomarker discovery. (a) Large retinal image dataset, (b) pre-processing, (c) no pre-processing, (d) arteriole/venule segmentation, (e) optic disc/cup segmentation, (f) image quality classification, (g) vessel analysis, (h) data exclusion, and (i) statistical modelling for biomarker discovery.

The adapted U-Net underwent pretraining via self-supervised learning using the EyePACS dataset from Kaggle [31], consisting of 88,702 retinal images. An autoencoder architecture was achieved by modifying the model to generate the reconstructed input image as its output, along with the removal of skip connections between the encoder and decoder to prevent early feature maps from the encoder being directly employed in image reconstruction. The encoder section of the autoencoder was then used as weights for the encoder section of the adapted U-Net.

For each of the three datasets, two human observers manually annotated 25 randomly selected retinal images. Random selection was repeated until a wide spectrum of image qualities was achieved, along with inclusion of cases with eye disease for the CLSA and NEL DESP datasets, replacing those deemed of inadequate quality by further random selection. Images were annotated as standard, with pixel labels of arteriole, venule, crossing (arteriole and venule overlap), uncertain (vessel undistinguishable as either arteriole or venule), and background. The annotations were then adapted so each channel was represented as a binary map with the positive classes of arterioles, venules, and vessels respectively. Crossing pixels belonged to all three positive classes, and uncertain pixels belonged only to the positive class in the binary map for vessels. Uncertain pixels were masked out of the binary maps for arterioles and venules to stop them being labelled as the negative class. The annotated images were divided with a random 60:20:20 training, validation, and test split.

The pre-processed images were resized to 1024 x 1024 pixels for input to the model, the output was the same size. Horizontal and vertical flipping, scaling, translation, rotation, brightness, contrast, and saturation were used to augment the training data. The batch size was 2. Adam optimization was used with a learning rate of 0.001 for 1000 epochs, learning rate decay presented no improvements. A weight decay of 0.005 was used. The loss function was pixel-wise binary cross-entropy summed across the channels. Weighting the loss function (e.g., sample weights) offered no improvements. The model was saved at the epoch with the minimum validation loss. The model was built on the training set and hyperparameters were derived from performance on the validation set.

The output probability maps were returned to the size prior to pre-processing and then thresholded to produce binary segmentation maps, followed by simple post-processing [5] to eliminate spurious objects. Morphological thinning was applied to the vessel binary map to create vessel centrelines, followed by removal of spurs, bifurcations, and crossover points to create vessel segments [5]. The scores from the arteriole and venule probability maps were used as soft votes and accumulated along the vessel centreline pixels to determine the AV probability for each vessel segment. Using the centrelines proved more effective than using the entire vessel segment.

### C. Optic Disc/Cup Segmentation

The adapted U-Net architecture in the A/V segmentation module was repurposed for optic disc and optic cup segmentation. The network outputs two independent channels to generate separate binary segmentation maps for the structures of optic disc (OD) and optic cup (OC). This configuration enables the model to understand that the entirety of the area belongs to the OD, whilst only the inner area belongs to the OC. The adapted U-Net was pretrained using the REFUGE dataset [32], consisting of 1200 annotated retinal images.

For each of the three datasets, two human observers manually annotated 100 randomly selected retinal images. The same selection procedure as detailed for the A/V segmentation module. Images were annotated as standard, with pixel labels of OD, OC, and background. The OC

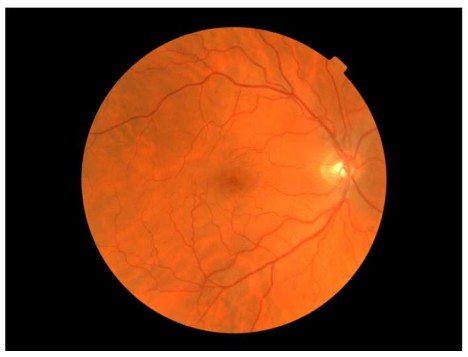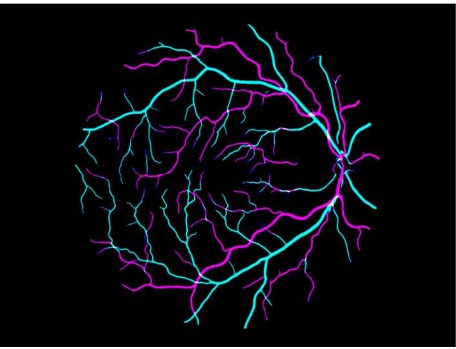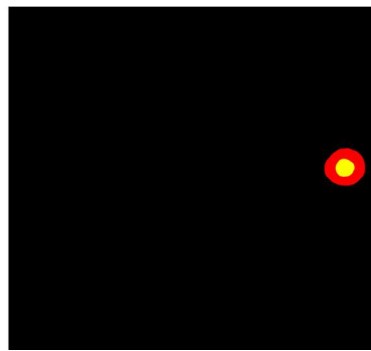

Fig. 2. QUARTZ-DL segmentation example for UKBB. Left: original retinal image. Middle: A/V segmentation, figure depicts the RGB composition of three independent binary segmentation maps for the structures of arterioles, venules, and vessels. Right: OC/OD segmentation, figure depicts the RGB composition of two independent binary segmentation maps for the structures of the OD and OC. ©UK Biobank.

annotation is an approximation based on the colour difference between the area of central pallor which contrasts to the pink/orange of the neuroretinal rim. Accurately defining the OC would require detailed contour information such as that available from optical coherence tomography. The annotations were then adapted so each channel was a binary map with the positive classes of OD and OC respectively. OC pixels belonged to both positive classes. The annotated images were divided with a random 60:20:20 training, validation, and test split.

The training strategy closely resembled that of the A/V segmentation module, with a few modifications: pre-processed images resized to 512 x 512 pixels, used a batch size of 8, trained for 100 epochs, and employed a cosine decay learning rate schedule.

The output probability maps were returned to the size prior to pre-processing and then thresholded to produce binary segmentation maps, with the largest segmented connected component determining the object of interest in each map. OD localization was determined from the centroid of the segmented OD. Additionally, the segmented OD and OC were used to calculate the vertical cup-to-disc ratio (vCDR) [32].

*D. Image Quality Classification*

Large retinal datasets used in epidemiological studies can contain large amounts of poorer quality images. However, useful information can be extracted from well segmented sections of the vasculature, even if this only represents a portion of the vascular tree. A dual CNN model employing two instances of the EfficientNetV2-S architecture [33] was created to evaluate image quality with respect to suitability for epidemiological studies. One network took A/V segmentation maps as input, while the other took colour retinal images as input. The feature maps generated by both networks were concatenated and subjected to global average pooling. This was followed by two fully connected layers of 256 nodes each and a binary classification layer for distinguishing the classes of inadequate and adequate. Each instance of the EfficientNetV2-S was pretrained on the ImageNet dataset, followed by further pretraining of the entire model using 28,792 labelled retinal images from the EyeQ dataset and their output from the A/V segmentation module.

For each of the three datasets, a human observer manually labelled 2000 randomly selected retinal images as either inadequate or adequate. Using both the A/V segmented maps and the retinal images, images were labelled as inadequate if they met any of the following criteria: (i) contained considerable blur, (ii) were non-retinal images, (iii) less than half of the vasculature was segmented, (iv) the segmentation was considerably fragmented or unconnected, (v) multiple non-vessel objects were segmented (e.g., false segmentation caused by eyelashes, lens artefacts, choroidal vessels, exudates, haemorrhages, the fovea, the optic disc, retinal scars, retinitis pigmentosa, asteroid hyalosis etc.). The UKBB dataset had 79.10% of images manually labelled as adequate, while the CLSA dataset had 82.90%. The NEL DESP dataset had the lowest proportion of adequate images at 63.15%, mainly due to the inclusion of many non-retinal images, this increased to 84.77% when considering only retinal images. The labelled images were divided with a random 80:10:10 training, validation, and test split.

Images were not pre-processed as determining the correct FOV location for cropping is not always viable for some low quality images. Images were resized to 384 x 384 pixels for input to the model. Data augmentation was consistent with other modules, except for excluding translation and scaling as these can alter the amount of retina captured which is an important indicator for quality. The batch size was 16. Adam optimization was used with a learning rate of 0.001, learning rate decay presented no improvements. A weight decay of 0.0001 was used. The loss function was binary cross-entropy. Weighting the loss function offered no improvements. The model was saved at the epoch with the minimum validation loss. The model was built on the training set and hyperparameters were derived from performance on the validation set. The pretrained model's three-class softmax classification layer for the EyeQ dataset was replaced with a sigmoid classification layer with randomly initialised weights. Only the top layers of the model were trained, initially 5 epochs for the classification layer, followed by 20 epochs for the rest of the top.

A threshold was applied to the output probability score to determine and exclude images of inadequate quality. Additionally, using the original image size prior to any resizing detailed in the Materials section, images failing to meet the dataset's predominant aspect ratio were excluded. Deviation from this aspect ratio was likely due to cropped images or images with odd capture settings, which then becomes problematic as it impedes the camera's pixel-to-micron conversion. Also, images with dimensions below 500 pixels were excluded to ensure a minimum resolution requirement.

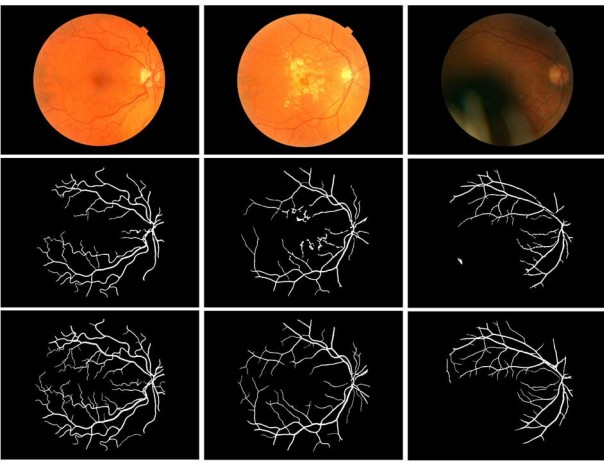

Fig. 3. Comparison of UKBB vessel segmentations. Top: original retinal image. Middle: vessel segmentation maps from QUARTZ [5]. Bottom: vessel segmentation maps from A/V segmentation for QUARTZ-DL. ©UK Biobank.

## IV. RESULTS

Evaluations were performed on the test sets. The operating point for each CNN model was determined by finding the probability threshold that maximised the F1 score on the validation set, which was the preferred metric due to the class imbalance for most of the tasks. Performance metrics include sensitivity, specificity, precision (positive predictive value), negative predictive value (NPV), accuracy, F1 score, intersection over union (IoU), area under the receiver operating characteristic curve (AUC ROC), and area under the precision-recall curve (AUC PR) [34]. In this section, the suffix -DL will be appended to QUARTZ to distinguish the newer deep learning version of the system.

The results of A/V segmentation are reported in Tables I-II. For QUARTZ-DL, the binary segmentation of each structure (arteriole, venule, and vessels) was evaluated separately. Only pixels within the circular FOV were used for evaluation. Further to this, uncertain pixels were excluded from the evaluation of arteriole and venule segmentation, but not for the vessel segmentation. The F1 score for vessel segmentation increases by 0.0719 from QUARTZ to QUARTZ-DL for UKBB, from 0.7753 to 0.8472. After post-processing, that F1 score increase is 0.0596, from 0.7724 to 0.8320. The introduction of post processing may reduce the F1 score but it's essential to keeping the segmentation of non-vessel objects to a minimum. Examples of outputs from A/V segmentation are provided in Fig. 2-3. The subsequent A/V segment-level performance, based on a 0.5 probability threshold, is reported in Table III. Accuracy increases by 0.0498 for the A/V segment-level decision from QUARTZ to QUARTZ-DL for UKBB, from 0.8524 to 0.9022. To further increase performance, epidemiological studies [3] have adopted a 0.8 threshold at the cost of only retaining 51.75% of the vessel segments, for QUARTZ-DL on UKBB this threshold now retains 73.58% of the vessel segments.

The results of OD/OC segmentation are reported in Table IV. For QUARTZ-DL, the binary segmentation of each structure (OD and OC) was evaluated separately. Only pixels within the circular FOV were used for evaluation. An output example of OD/OC segmentation is provided in Fig. 2. The detection rate for OD localization, evaluated over a further 300 test images, is reported in Table VI. The detection rate for

OD localization increases by 0.0173 from QUARTZ to QUARTZ-DL for UKBB, from 0.9760 to 0.9933. The mean absolute error (MAE) for vCDR is reported in Table VI.

The F1 score for image quality classification (Table V) from QUARTZ to QUARTZ-DL for UKBB increases by 0.0878, from 0.8872 to 0.9750. It is noteworthy that the retinal images in Fig. 3 were all classified as adequate quality, with adequate segmentations achieved despite the presence of heavily visible ocular disease and poor illumination.

The reported results from other retinal vasculometry systems offering a generalised approach [21], [22] (compared to the dataset specific versions of QUARTZ and QUARTZ-DL) have been incorporated within Tables I, IV, and V. Reported results from [21] include evaluation on external datasets and multiclass performance for A/V segmentation and OD/OC segmentation. However, direct comparison by utilizing datasets in this paper was hindered because numerous test images were deemed as insufficient quality by those systems, resulting in no outputs being provided. Instead, for each segmentation task, a UKBB test subset using only images deemed as sufficient quality by AutoMorph [21] was used for comparison. To aid comparison, binary segmentation maps for each structure (arteriole, venule, OD, and OC) were extracted from the AutoMorph multiclass segmentation maps, while the vessel map came from the binary vessel segmentation module.

The results from processing the entire datasets are detailed in Table VII, reporting the numbers of images that have useful information extracted for use in epidemiological studies. Participant consent withdrawals result in a small reduction in the number of UKBB retinal images. The number of images labelled as adequate quality for UKBB from QUARTZ to QUARTZ-DL increases by 7.77 percentage points, from 67.50% to 75.27%. Not meeting the dataset's predominant aspect ratio was only an issue for NEL DESP, accounting for 7.68% of the dataset. A range of Nvidia graphics processing units (GPUs) was used including the RTX 3080 Ti, RTX 4090, and A100. The average processing time for a single image on a standard machine (i9 3.60GHz, RTX 3080 Ti) was 5, 12, and 21 seconds for UKBB, NEL DESP, and CLSA, respectively.

## V. DISCUSSION AND CONCLUSION

QUARTZ is a robust fully automated artificial intelligence-enabled retinal vasculometry system which can process large retinal datasets to obtain quantitative measures of vessel morphology for use in epidemiological studies which have shown that these measures can be used in risk prediction models with application to population screening, particularly for circulatory disease, stroke, and coronary heart disease. In this paper, the latest version of QUARTZ (i.e., QUARTZ-DL) is evaluated, presenting the performance increases from shifting to a deep learning pipeline. These increases include +0.0719 to the F1 score for vessel segmentation, +0.0498 to accuracy for the A/V segment-level decision, +0.0173 to the detection rate for OD localization, and +0.0878 to the F1 score for image quality classification. The deep learning pipeline ensures that the previous two modules of vessel segmentation and A/V classification can be streamlined into a single A/V segmentation module. Also, the new addition of OD/OC segmentation, enables the new metric of vCDR to be calculated. The enhancements made to QUARTZ will result

TABLE I: Performance of A/V Segmentation.

| Method | Map | Sensitivity | Specificity | Precision | Accuracy | F$_1$ score | IoU | AUC ROC | AUC PR |
|---|---|---|---|---|---|---|---|---|---|
| RMHAS [22] (In-house) | Arteriole | 0.72 | 0.96 | - | 0.95 | 0.48 | - | 0.94 | - |
| | Venule | 0.80 | 0.97 | - | 0.96 | 0.57 | - | 0.96 | - |
| AutoMorph [21] (IOSTAR-AV) | A/V | 0.64 | 0.98 | 0.68 | 0.96 | 0.66 | 0.53 | 0.95 | - |
| AutoMorph [21] (DR HAGIS) | Vessel | 0.84 | 0.98 | 0.73 | 0.97 | 0.78 | 0.64 | 0.98 | - |
| AutoMorph (UKBB subset) | Arteriole | 0.6198 | 0.9916 | 0.7871 | 0.9738 | 0.6935 | 0.5308 | - | - |
| | Venule | 0.6813 | 0.9920 | 0.8269 | 0.9754 | 0.7471 | 0.5963 | - | - |
| AutoMorph (UKBB subset) | Vessel | 0.7688 | 0.9895 | 0.8954 | 0.9665 | 0.8273 | 0.7054 | - | - |
| QUARTZ [5] (UKBB) | Vessel | 0.7366 | 0.9814 | 0.8183 | 0.9564 | 0.7753 | 0.6330 | - | - |
| QUARTZ-DL (UKBB) | Arteriole | 0.7699 | 0.9875 | 0.7526 | 0.9773 | 0.7612 | 0.6144 | 0.9866 | 0.8479 |
| | Venule | 0.7832 | 0.9902 | 0.8144 | 0.9794 | 0.7985 | 0.6645 | 0.9901 | 0.8895 |
| | Vessel | 0.8545 | 0.9814 | 0.8401 | 0.9684 | 0.8472 | 0.7349 | 0.9896 | 0.9313 |
| QUARTZ-DL (UKBB subset) | Arteriole | 0.7730 | 0.9869 | 0.7484 | 0.9767 | 0.7605 | 0.6135 | - | - |
| | Venule | 0.7823 | 0.9895 | 0.8076 | 0.9785 | 0.7947 | 0.6594 | - | - |
| | Vessel | 0.8577 | 0.9807 | 0.8380 | 0.9678 | 0.8478 | 0.7357 | - | - |
| QUARTZ-DL (CLSA) | Arteriole | 0.7757 | 0.9872 | 0.7642 | 0.9764 | 0.7699 | 0.6259 | 0.9859 | 0.8499 |
| | Venule | 0.7976 | 0.9894 | 0.8187 | 0.9785 | 0.8080 | 0.6779 | 0.9893 | 0.8961 |
| | Vessel | 0.8581 | 0.9803 | 0.8445 | 0.9667 | 0.8513 | 0.7410 | 0.9883 | 0.9312 |
| QUARTZ-DL (NEL DESP) | Arteriole | 0.7627 | 0.9905 | 0.7861 | 0.9806 | 0.7742 | 0.6316 | 0.9878 | 0.8585 |
| | Venule | 0.8046 | 0.9913 | 0.8262 | 0.9821 | 0.8153 | 0.6881 | 0.9908 | 0.8966 |
| | Vessel | 0.8462 | 0.9836 | 0.8471 | 0.9703 | 0.8467 | 0.7341 | 0.9896 | 0.9309 |

TABLE II: Performance of A/V Segmentation After Post-processing.

| Method | Map | Sensitivity | Specificity | Precision | Accuracy | F$_1$ score | IoU |
|---|---|---|---|---|---|---|---|
| QUARTZ [5] (UKBB) | Vessel | 0.6912 | 0.9888 | 0.8752 | 0.9584 | 0.7724 | 0.6292 |
| QUARTZ-DL (UKBB) | Arteriole | 0.7506 | 0.9890 | 0.7710 | 0.9778 | 0.7607 | 0.6138 |
| | Venule | 0.7697 | 0.9910 | 0.8249 | 0.9795 | 0.7963 | 0.6616 |
| | Vessel | 0.8112 | 0.9841 | 0.8539 | 0.9664 | 0.8320 | 0.7123 |
| QUARTZ-DL (CLSA) | Arteriole | 0.7496 | 0.9888 | 0.7814 | 0.9766 | 0.7652 | 0.6197 |
| | Venule | 0.7897 | 0.9901 | 0.8275 | 0.9788 | 0.8082 | 0.6781 |
| | Vessel | 0.8357 | 0.9823 | 0.8551 | 0.9661 | 0.8453 | 0.7321 |
| QUARTZ-DL (NEL DESP) | Arteriole | 0.7372 | 0.9921 | 0.8099 | 0.9810 | 0.7718 | 0.6285 |
| | Venule | 0.7947 | 0.9920 | 0.8364 | 0.9823 | 0.8150 | 0.6878 |
| | Vessel | 0.8198 | 0.9864 | 0.8657 | 0.9702 | 0.8421 | 0.7273 |

TABLE III: Performance of A/V Segment-Level Decision.

| Method | Class | Sensitivity | Specificity | Precision | Accuracy | F$_1$ score |
|---|---|---|---|---|---|---|
| QUARTZ [5] (UKBB) | Arteriole | 0.8514 | 0.8532 | 0.8123 | 0.8524 | 0.8314 |
| | Venule | 0.8532 | 0.8514 | 0.8849 | 0.8524 | 0.8688 |
| QUARTZ-DL (UKBB) | Arteriole | 0.9079 | 0.8971 | 0.8857 | 0.9022 | 0.8967 |
| | Venule | 0.8971 | 0.9079 | 0.9173 | 0.9022 | 0.9071 |
| QUARTZ-DL (CLSA) | Arteriole | 0.8804 | 0.8713 | 0.8567 | 0.8755 | 0.8684 |
| | Venule | 0.8713 | 0.8804 | 0.8929 | 0.8755 | 0.8820 |
| QUARTZ-DL (NEL DESP) | Arteriole | 0.8882 | 0.9290 | 0.9073 | 0.9111 | 0.8977 |
| | Venule | 0.9290 | 0.8882 | 0.9140 | 0.9111 | 0.9214 |

TABLE IV: Performance of OD/OC Segmentation.

| Method | Map | Sensitivity | Specificity | Precision | Accuracy | F$_1$ score | IoU | AUC ROC | AUC PR |
|---|---|---|---|---|---|---|---|---|---|
| AutoMorph [21] (IDRID) | OD/OC | 0.90 | 0.95 | 0.94 | 0.99 | 0.94 | 0.91 | 0.95 | - |
| AutoMorph (UKBB subset) | OD | 0.8367 | 0.9993 | 0.9618 | 0.9958 | 0.8949 | 0.8098 | - | - |
| | OC | 0.8425 | 0.9988 | 0.7225 | 0.9982 | 0.7779 | 0.6365 | - | - |
| QUARTZ-DL (UKBB) | OD | 0.9568 | 0.9991 | 0.9571 | 0.9982 | 0.9569 | 0.9174 | 0.9999 | 0.9945 |
| | OC | 0.8746 | 0.9993 | 0.8209 | 0.9988 | 0.8469 | 0.7344 | 0.9997 | 0.9318 |
| QUARTZ-DL (UKBB subset) | OD | 0.9464 | 0.9991 | 0.9603 | 0.9980 | 0.9533 | 0.9108 | - | - |
| | OC | 0.8819 | 0.9993 | 0.8255 | 0.9988 | 0.8527 | 0.7433 | - | - |
| QUARTZ-DL (CLSA) | OD | 0.9704 | 0.9992 | 0.9632 | 0.9985 | 0.9668 | 0.9358 | 0.9999 | 0.9965 |
| | OC | 0.8873 | 0.9993 | 0.8414 | 0.9989 | 0.8640 | 0.7606 | 0.9998 | 0.9471 |
| QUARTZ-DL (NEL DESP) | OD | 0.9582 | 0.9984 | 0.9289 | 0.9975 | 0.9433 | 0.8927 | 0.9997 | 0.9880 |
| | OC | 0.8254 | 0.9991 | 0.8374 | 0.9982 | 0.8313 | 0.7114 | 0.9994 | 0.9215 |

TABLE V: Performance of Image Quality Classification, Positive Class = Inadequate.

| Method | Sensitivity | Specificity | Precision | NPV | Accuracy | F$_1$ score | AUC ROC | AUC PR |
|---|---|---|---|---|---|---|---|---|
| AutoMorph [21] (EyeQ) | 0.85 | 0.93 | 0.87 | - | 0.92 | 0.86 | 0.97 | - |
| QUARTZ [5] (UKBB) | 0.9500 | 0.9395 | 0.8321 | 0.9835 | 0.9420 | 0.8872 | 0.9679 | - |
| QUARTZ-DL (UKBB) | 0.9512 | 1.0000 | 1.0000 | 0.9876 | 0.9900 | 0.9750 | 0.9974 | 0.9922 |
| QUARTZ-DL (CLSA) | 0.9118 | 1.0000 | 1.0000 | 0.9822 | 0.9850 | 0.9538 | 0.9945 | 0.9793 |
| QUARTZ-DL (NEL DESP) | 0.9737 | 0.9919 | 0.9867 | 0.9840 | 0.9850 | 0.9801 | 0.9950 | 0.9937 |

TABLE VI: OD LOCALIZATION AND THE MAE FOR vCDR.

| Method | Detection rate | MAE for vCDR |
|---|---|---|
| QUARTZ [5] (UKBB) | 0.9760 | - |
| QUARTZ-DL (UKBB) | 0.9933 | 0.0560 |
| QUARTZ-DL (CLSA) | 0.9900 | 0.0378 |
| QUARTZ-DL (NEL DESP) | 0.9900 | 0.0651 |

TABLE VII: PROCESSING ENTIRE DATASETS.

| Method | Processed | Adequate image quality |
|---|---|---|
| QUARTZ [5] (UKBB) | 135,867 | 97,188 (71.53%) |
| QUARTZ (UKBB) | 175,856 | 118,702 (67.50%) |
| QUARTZ-DL (UKBB) | 175,764 | 132,293 (75.27%) |
| QUARTZ-DL (CLSA) | 106,506 | 88,155 (82.77%) |
| QUARTZ-DL (NEL DESP) | 1,175,423 | 803,555 (68.36%) |

in higher-quality retinal vasculometry data, thereby contributing to the improvement of future epidemiological studies.

QUARTZ's deep learning modules demonstrate performance consistent with other retinal vasculometry systems that use a deep learning pipeline, outperforming AutoMorph [21] on the UKBB dataset. However, the focus was not on direct comparison since [21], [22] are generalised systems offering broader application. Instead, QUARTZ provides a contribution in terms of its efficient use of data, extracting information from well segmented sections of the vasculature, even if this only represents a portion of the vascular tree. This is important when dealing with large cohorts used in epidemiological studies which often include larger amounts of poorer quality images including partially illuminated images. A potential impact of the approach to include the contribution of partially illuminated retinal images is that different areas of the retina have a different vascular morphology. However, the requirement that at least half of the vasculature must result in a high-quality segmentation ensures that a substantial portion of the retinal vessels can be reliably represented and this criterion contributes towards mitigating this impact. All the deep learning modules in QUARTZ have been trained and evaluated on a diverse range of images. On the image quality test set of UKBB, AutoMorph labels 57.50% of the images as adequate quality compared to 80.50% labelled by QUARTZ (75.27% on the entire dataset).

The high performance of QUARTZ achieved on UKBB holds for the two other large datasets used in this study, CLSA and NEL DESP, demonstrating the robustness of its modules across datasets. In the case of the NEL DESP, QUARTZ can remove retinal images of inadequate quality, as well as non-retinal images which are present in all real-world large datasets. Removal of non-retinal images is normally not covered in the literature. Segmentation via deep learning, in addition to the ocular disease present in the CLSA and NEL DESP datasets (with the NEL DESP dataset containing ~12% of patients with referable diabetic retinopathy), enables QUARTZ to learn to better avoid false segmentations in the presence of ocular disease. In addition, QUARTZ demonstrates a decrease in false segmentations, addressing various factors such as eyelashes, lens artefacts, and choroidal vessels, among others. The choroid thins with age, making choroidal vessels more prominent [35]. Despite QUARTZ improvements, in severe cases, these vessels can be mistaken for retinal vessels, resulting in occasional incorrect segmentations and hindering quality classification. Consequently, the F1 score in image quality classification for

the CLSA dataset is lower compared with the other two datasets, despite remaining notably high at 0.9538.

QUARTZ achieves a very high OD localization detection rate due to high performing OD segmentation on all three datasets. This enables the differentiation of left from right eye for macular centred images, as well knowing the location of a vessel relative to the OD. On the other hand, as our current goal is to maximise the usage of segmented vessels, OC segmentation and subsequently vCDR are hampered by the many poorly illuminated ODs. Using the vCDR in epidemiological studies requires a separate image quality classification decision based on the clarity of the OD. The adapted U-Net architecture when trained and evaluated on the REFUGE dataset, which only contains high quality images, is in line with the state-of-the-art with a MAE for vCDR of 0.0330.

Following evaluation, QUARTZ was used to automatically process multiple entire retinal datasets. A high percentage of images, 75.27% for UKBB and 82.77% for CLSA, being deemed as adequate for use as vasculometry data in epidemiological studies. This represents a large improvement from the previous version of QUARTZ, due to both improved vessel segmentation and image quality classification. For NEL DESP, the highest percentage would be expected with respect to the quality of the retinal images, instead 68.36% is achieved which reflects the removal of non-retinal images and images that don't comply to the dataset's predominant aspect ratio. A retinal vasculometry system processing over a million images from a single dataset is a significant milestone, which has been achieved in the NEL DESP dataset. The entire NEL DESP dataset was processed in 2 months using a system with an AMD 3.65GHz processor and 4 x A100 GPUs, making the practical application of this system to large scale studies feasible.

Future work will involve the transition to a generic version of QUARTZ, expanding its utility to a broader audience. This generalised system will undergo training using multiple datasets, encompassing diverse demographics, a spectrum of diseases, and various camera setups. The advantages and disadvantages will be explored, considering whether performance might be compromised compared with a system tailored to a single population, as well as whether the increased diversity and number of training examples would benefit individual populations.

In conclusion, QUARTZ has transitioned to a deep learning pipeline, demonstrating improvements of its modules, and high performance across datasets. QUARTZ has been successfully used to process large datasets, extracting information from a high percentage of retinal images. These retinal vasculometry outputs will serve as a valuable resource for epidemiological studies.

## COLLABORATORS

The UK Biobank Eye and Vision Consortium, the list of members is available from the consortium website (http://www.ukbiobankeyeconsortium.org.uk/people).

The Artificial Intelligence & Automated Retinal Image Analysis Systems (ARIAS) Research Group: John Anderson, Sarah Barman, Louis Bolter, Ryan Chambers, Lakshmi Chandrasekaran, Umar Chaudhry, Emily Y Chew, Catherine Egan, Jiri Fajtl, Frederick L Ferris, Aroon D Hingorani, Aaron

Y Lee, Abraham Olvera-Barrios, Christopher G Owen, Paolo Remagnino, Alicja R Rudnicka, Royce Shakespeare, Reecha Sofat, Adnan Tufail, Alasdair N Warwick, Charlotte Wahlich, Roshan Welikala, Kathryn Willis, Yue Wu.

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
