# OpenReview forum: "Artificial Intelligence-Enabled Retinal Vasculometry at Scale Utilizing the UK Biobank, CLSA, and NEL DESP Datasets"
_IEEE.org/EMBS/BHI/2024/Conference — IEEE BHI'24_

### Official Review · Reviewer_qEhi · 2024-08-06
**Artificial Intelligence-Enabled Retinal Vasculometry at Scale Utilizing the UK Biobank, CLSA, and NEL DESP Datasets**

**Overall Rating:** 7
**Confidence:** 5

**Other Quality Metrics:**

excellent

**Questions For The Authors:**

Can you provide more details about the strutured data base adequate for your study?

**Strengths:**

The study is well conducted and in an important issue

**Summary Of The Paper:**

Non invasive methods are the future for medical procedures and for patients well fare.

**Weaknesses:**

Nothing to say negative

---

### Official Review · Reviewer_vQ1w · 2024-08-10
**Review AI-Enabled Retinal Vasculometry**

**Overall Rating:** 8
**Confidence:** 4

**Other Quality Metrics:**

- (a) Clarity of writing: Excellent
- (b) Clinical Significance: Excellent
- (c) Methodological Novelty: Good
- (d) Experiments and Results: Great

**Questions For The Authors:**

- I understand that the authors state that the focus of the current article was not to compare their methods against other methods on the same tasks (arteriole/venule segmentation, optic disc/cup segmentation, image quality classification, and vessel analysis) but would it be possible to employ the other methods from the literature (Automorph) on the same data as was used in the current article (results tables)? To better understand how it compares against them.

- From my understanding, in the different tasks considered, the models were pre-trained on external datasets and then trained on the datasets used in the current work on a small subset of images manually annotated by experts. Was that it? I believe that the authors should clearly state the number of samples considered in the training process, for instance, together with the data spilled ratio, to clarify it.

**Strengths:**

- Overall, the article is very well laid out and written. The ideas are clear and well organized, with a logical sequence

- The use of large-scale databases from multiple sources demonstrates the generalization capability of the developed models

- The potential use cases of the developed models are clear, as well as the impact on real-world applications

- Models demonstrate good overall performance in the different tasks considered. Improvements over the previous version of the models are also a strong point

**Summary Of The Paper:**

In this article, the authors propose a fully automated artificial intelligence-enabled retinal vasculometry system. Their system has transitioned from a previous version based on traditional signal processing techniques to a deep learning pipeline, demonstrating improved performance metrics in processing large-scale retinal image datasets for epidemiological studies.

**Weaknesses:**

- Perhaps in section I (Introduction) the authors could better highlight the contributions/advancements of the current work and present the method used in the other works in the literature (VAMPIRE, AutoMorph, RMHAS)

---

### Official Review · Reviewer_32Mq · 2024-08-18
**Artificial Intelligence-Enabled Retinal Vasculometry at Scale Utilizing the UK Biobank, CLSA, and NEL DESP Datasets**

**Overall Rating:** 7
**Confidence:** 3

**Other Quality Metrics:**

(a) Clarity of writing; 	Good
(b) Clinical Significance; 	Great
(c) Methodological Novelty; 	Great
(d) Experiments and Results	Good

**Questions For The Authors:**

Since many part of text is hidden, the reading of this paper was quiet cumborsome. At many places it is mentioned improved version. does it means that the work is an improvement of authros own other work?

**Strengths:**

The ability to handle images with low illumination or those affected by ocular diseases is a notable advancement.
The processing of images from the NEL DESP dataset demonstrates the system's scalability and its potential for practical application in large-scale epidemiological studies.

**Summary Of The Paper:**

The paper presents an AI-enabled system for analysing retinal images to extract quantitative measures of blood vessel morphology
The system utilizes deep learning models for core tasks such as blood vessel segmentation, optic disc/cup segmentation, and image quality classification. The system has been successfully applied to large datasets like the UK Biobank, CLSA, and NEL DESP, showcasing its ability to process vast amounts of data efficiently. The extracted retinal vasculometry data is expected to be a valuable resource for future epidemiological studies.

**Weaknesses:**

While the authors mention other retinal vasculometry systems like AutoMorph and RHMAS, the direct comparison is limited due to the inability of those systems to process many test images deemed of insufficient quality.

A more comprehensive comparison, perhaps by adapting those systems or using a subset of images that meet their quality criteria, could provide a clearer picture of proposed work's relative strengths and weaknesses

While the system's performance across different datasets is promising, external validation on independent datasets would further strengthen the claims of its robustness and generalizability

---

### Decision · Program_Chairs · 2024-09-23

Accept